# Centrality measures in psychological networks: A simulation study on identifying effective treatment targets

Daniel Castro[1,2]*, Deisy Gysi[3], Filipa Ferreira[1,2], Fernando Ferreira-Santos[4], Tiago Bento Ferreira[1,2]

**1** University of Maia, Maia, Portugal, **2** Center for Psychology at University of Porto, Porto, Portugal, **3** Center for Complex Network Research, Northeastern University, Boston, Massachusetts, United States of America, **4** Laboratory of Neuropsychophysiology, Faculty of Psychology and Education Sciences, University of Porto, Porto, Portugal

\* dcastro@umaia.pt

**Data Availability Statement:** The data underlying the results presented in the study are available from https://osf.io/mgdp6 or https://osf.io/5ybxt/.

## Abstract

The network theory of psychopathology suggests that symptoms in a disorder form a network and that identifying central symptoms within this network might be important for an effective and personalized treatment. However, recent evidence has been inconclusive. We analyzed contemporaneous idiographic networks of depression and anxiety symptoms. Two approaches were compared: a cascade-based attack where symptoms were deactivated in decreasing centrality order, and a normal attack where symptoms were deactivated based on original centrality estimates. Results showed that centrality measures significantly affected the attack's magnitude, particularly the number of components and average path length in both normal and cascade attacks. Degree centrality consistently had the highest impact on the network properties. This study emphasizes the importance of considering centrality measures when identifying treatment targets in psychological networks. Further research is needed to better understand the causal relationships and predictive capabilities of centrality measures in personalized treatments for mental disorders.

## Introduction

In recent years, the field of psychology has increasingly acknowledged the necessity of personalized treatments [1–3]. This recognition has been facilitated by advancements in technology and longitudinal assessment methodologies [4]. Within this context, network analysis emerged as one of the most promising methodological approaches to study this type of data [5–7]. By modeling mental disorders as a network of symptoms, where symptoms are viewed as nodes and connections between them as edges [8]. From this methodological approach, the network theory of psychopathology emerged [8–11]. The network theory of psychopathology [9, 10] proposes that when a symptom is activated (such as by an external event) a signal diffuses through the network, activating other symptoms. The activation of other symptoms increases the network connectivity and the system transitions into a disease state. Thus, a symptom with

**Funding:** This work was supported by national funding from the Portuguese Foundation for Science and Technology (UIDB/00050/2020). DC is supported by the Portuguese Foundation for Science and Technology through the Ph.D. grant: SFRH/BD/148884/2019. The funders had no role in the study design, data collection and analysis, decision to publish, or preparation of the manuscript. There was no additional external funding received for this study.

**Competing interests:** The authors have declared that no competing interests exist.

more connections might activate several other symptoms and might have an essential role in sustaining the disease. Due to this, these symptoms have been suggested to be preferential treatment targets [8, 12].

In network analysis the identification of these symptoms can be performed through the estimation of centrality measures, such as degree, strength, betweenness, and closeness, which uncover each symptoms' connectivity. With these measures, and with the proposal from the network theory of psychopathology, several studies suggested possible treatment targets based on network centrality measures [13–16]. However, recent evidence showed that closeness and betweenness centrality are not adequate in psychopathological networks, given their bias in the covariance and sampling variability [17, 18]. Dablander and Hinne [19] shown that the most common centrality measures used in psychology are not related to causality, except for eigenvector centrality. This is partially validated by the inconsistent results shown by studies examining the central symptoms as psychotherapeutic targets [20–23].

Studies found central symptoms to predict changes in the remaining symptoms [24, 25] and enable the evolution to a psychopathological condition [26]. However, Bos and colleagues [20] have not found evidence to support the hypothesis that symptom centrality is associated with changes in symptoms over time across cross-sectional networks. Furthermore, three other studies used the same procedure in three different samples finding only moderate support for this hypothesis. Rodebaugh and colleagues [21] explored if a cross-sectional network of social anxiety symptoms predicted changes in another sample of individuals who undertook treatment for the same disorder. The authors concluded that symptom centrality was not generalized across measures and frequency of symptom endorsement also predicted change while being generalized across measures. Spiller and colleagues [22] and Papini and colleagues [23] have also concluded that symptom endorsement was a better predictor of change than the centrality measures, with only expected influence predicting how changes in symptoms were associated with changes in the remainder of the symptoms. In fact, deactivating symptoms according to their centrality does not seem to significantly reduce network density more than the random deactivation of symptoms [27].

Despite this, and in line with the studies that reported changes in the network structure when comparing psychological networks at different stages [28–33], when symptoms are deactivated according to their degree centrality there are significant changes in the number of components of the network [27]. However, most of these studies were performed using cross-sectional networks [20–23, 29, 30] which might have hampered their findings, due to individuals' idiosyncrasy that is lost in the cross-sectional analysis [34].

As a potential framework to unveil the individual-level differences one can leverage idiographic networks [35], which allow for a detailed understanding of the associations between symptoms, their directionality, and how different processes (e.g., thoughts, feelings, and behaviors) fluctuate over time [36]. It is based on this dynamicity that we can understand the diversity of clinical and symptomatologic trajectories. This idiographic dynamic view, coupled with the hypothesis of central symptoms being more efficient treatment targets, might promote the development of better personalized treatments. In fact, Levinson, and colleagues [37] reported initial evidence that identifying treatment targets through strength centrality in idiographic networks might improve treatments efficacy. However, Levinson and colleagues [37] only identified treatment targets at a single point (i.e., third session), which disregards the changes in the network that might occur after that point. It is expected that after an intervention directed to a symptom, changes in the network structure occur [8]. If these changes occur, a new central symptom may emerge, which should become the primary treatment target [27]. Failing to account for this possibility may result in an intervention that does not target the most central symptom, potentially reducing the efficacy of the treatment.

Considering this, the hypothesis that central symptoms might be valuable therapeutic targets remains open. Here we address this hypothesis by assessing the impact of symptom deactivation on idiographic networks according to different centrality measures. We do it, by comparing the impact of symptom deactivation using two different types of symptom deactivation procedures, one procedure based only on a single point estimate of centrality measures and a second procedure in which the centrality measures are re-estimated every time a symptom is deactivated. As usual in network science, the impact of symptom deactivation according to the centrality estimates is evaluated in comparison to the random deactivation of symptoms.

## Method

### Data

This study involved an analysis of existing data rather than new data collection. Datasets analyzed in the current study are publicly available (available at https://osf.io/mgdp6). Data was accessed in 9 of January of 2019 the authors did not had access to information that could identify individual participants. The ethics committee approval, the accordance of all the methods with relevant guidelines and regulations, and the informed consent were obtained in a previous published study by different authors [6]. The authors of this manuscript had no control over these data collection procedures. All the code used in this study is available at https://osf.io/k2z84/.

We analyze contemporaneous idiographic networks of depression and anxiety symptoms from 40 participants [6], which, in the original study, aimed to explore the idiographic structure of mood and anxiety symptomatology via contemporaneous and temporal networks. The participants had to have a diagnosis of major depressive disorder (MDD) or generalized anxiety disorder (GAD) and an age of 18 to 65 years, participants with a history of psychosis or mania were excluded. Most of the participants were female (65%), and 25 participants met the criteria for current GAD and 15 for current MDD (all the demographic data made available by the original study is presented in the S1 Appendix), network basic features are presented in S2 Table in S1 Appendix. Participants rated their symptoms for 30 days, four times per day through an Experience Sample Survey (ESS), which we used here to estimate temporal and contemporaneous networks of MDD and GAD symptoms for each participant. The ESS consisted of DSM-5 symptoms for MDD and GAD, where each participant rated their experience of each symptom in the preceding hours on a 0–100 scale and provided a mean (M) of 130.43 observations with a standard deviation (SD) of 19.27.

Here we focus on the contemporaneous networks. Contemporaneous networks can identify connections that would not be visible in temporal networks. This happens because temporal networks model the relationships that are predicted from one window of measurement to the next [38], which are, usually, in an interval of a few hours [39–41]. However, it is possible for causal relationships between variables to occur within timeframes that differ from those used to assess them but that can be identified through contemporaneous networks [38]. Thus, within-person contemporaneous networks might provide better identification of treatment targets. Therefore, here we focus on within-person contemporaneous networks to identify psychotherapeutic targets using centrality measures.

### Data analyses

For each participant, in the original study [6], a contemporaneous correlation matrix was extracted and then a sparse partial correlation network was estimated using the Least Absolute Shrinkage and Selection Operator (LASSO) regularization method [18]. On the original study

[6], model fit was assessed with RMSEA, Browns chi-square goodness-of-fit test, and the CFI. The authors considered non-significant chi-square tests, RMSEA values less than .060 and CFI values equal or greater than .95 to reflect an excellent fit. All participants exhibited an excellent fit on both chi-square and CFI. On RMSEA only one participant had a value of 0.062, with all the other participants exhibiting values below 0.060.

To identify the central nodes and their effect on individual networks, we perform a two-step analysis: i) the identification of central symptoms and ii) the exploration of the differential impact of symptoms' deactivation in the network. We investigate symptom deactivation as an indirect measure of symptom improvement, operating under the assumption that a symptom's complete recovery would result in its removal from the network of symptoms. This is done to simulate the effect of detecting a central symptom and then acting on it and improving it until the symptom is not felt / reported by participants.

The network exploration of symptom deactivation is performed as a cascade-based attack and a normal attack. In a cascade-based attack, symptoms are deactivated in their decreasing order according to their centrality, which is iteratively calculated at every symptom removal. In normal attack [27], symptoms were deactivated according to their original centrality. We compare this with random attack symptom deactivation procedure, where symptoms are randomly deactivated. In each type of attack a symptom is identified as a treatment target and deactivated from the network. For the cascade-based attack, symptom networks and treatment target selection are constantly being estimated and selected after each symptom deactivation. In the normal attack the treatment targets order is determined based on the initial symptom network of the participant and the attack follows that order without estimating the network again. Fig 1 illustrates an example of each attack and their differential impact on the average path length.

In the cascade-based attack and in the normal attack procedures' we identified the central symptoms through five different centrality measures: strength centrality, degree centrality, one-step and two-step expected influence centrality, and eigenvector centrality. Strength centrality, one-step and two-step expected influence were chosen due to their extensive use in psychopathological networks [42–44]. In addition, degree and eigenvector were selected due to being suggested as alternatives for better identification of treatment targets [19, 27]. After each symptom deactivation, a set of network properties was measured (i.e., network density, number of components, and average path length). The network density is the ratio between the number of edges in the network and all the potential edges [45], while the number of components in the network refers to the number of symptoms or groups of symptoms that are disconnected from the rest of the network and might be able to help us with the identification of groups of active symptoms. Finally, the average path length concerns the mean of the shortest paths in the networks and might identify if network symptoms activation is more likely due to the shortest distance between network symptoms.

For each centrality measure, we assessed the differential impact of symptom deactivation by computing the magnitude and extent of the attack in the network, where the impact magnitude consists of the difference between the maximum values of average path length and components and their initial values. We next measured the attack extent by computing the proportion of nodes that needed to be deactivated to achieve the maximum value of average path length and number of components. To assess the impact of symptom deactivation on the network density we computed the density of the network at 50% of the symptoms deactivated. All these analyses were performed in the package *psychNetsAttack* [46] for R [47]. We did this to each of the 40 networks (S3–S83 Figs in the S1 Appendix) and then aggregated the results from each of the 40 networks to compare the results of each centrality measure.

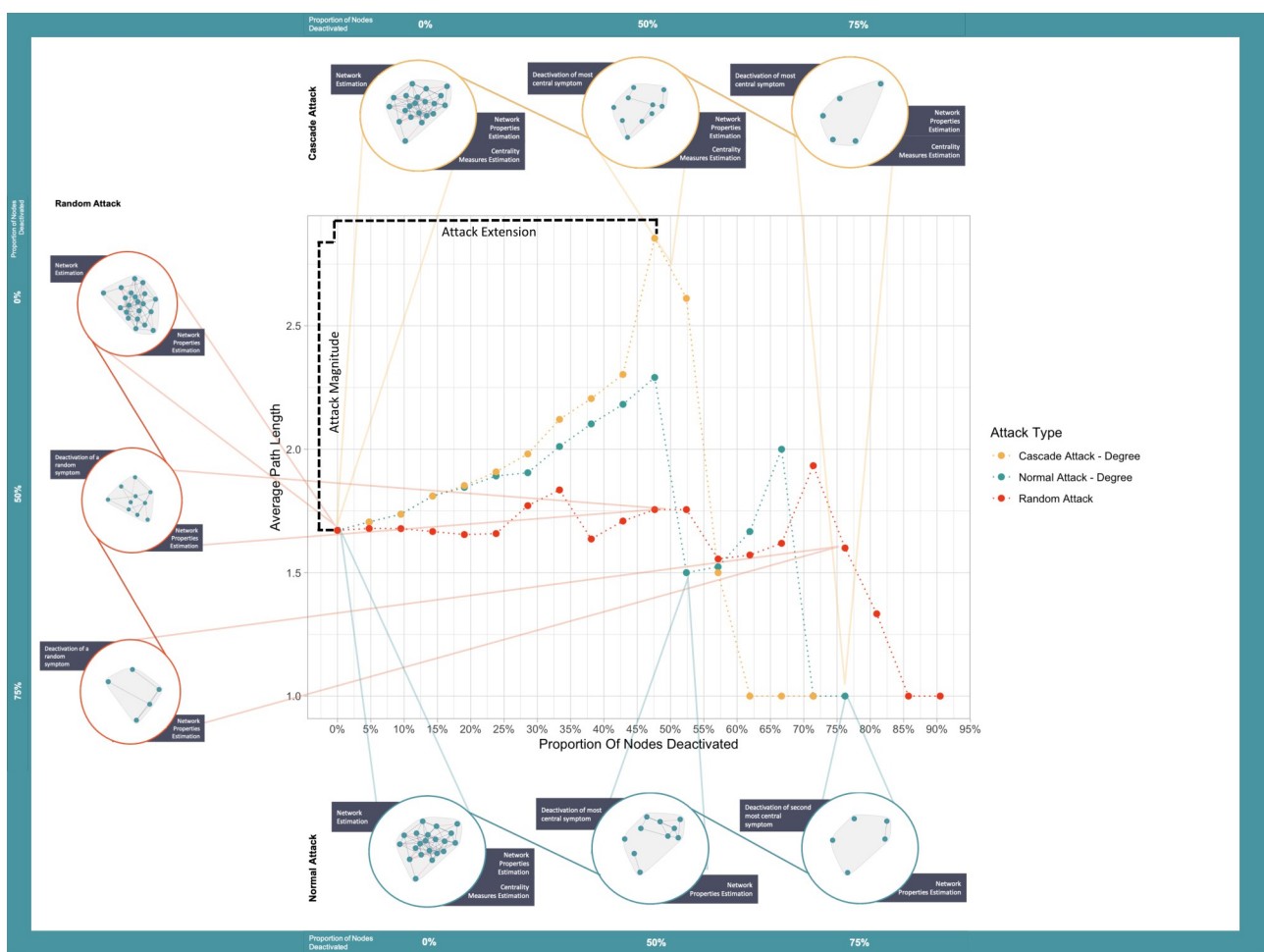

**Fig 1. Example representation of the 3 types of attacks and the 2 outcome measures.** An example representation of the 3 types of attacks (random attack and degree-based normal and cascade attack) and the 2 outcome measures (attack magnitude and attack extension). Attack methods are exemplified by the colored circles, yellow circles represent the cascade attack, orange circles the random attack, and green circles the normal attack. Colored dots represent the average path length of the network after the deactivation of the symptom identified by the attack. The X-axis represents the proportion of nodes deactivated and the Y-axis represents the average path length after each node deactivation.

To compare each centrality measure attack magnitude and extent on the number of components, average path length, and network density for both normal and cascade attack we used a Friedmann's Test, we followed by a Kendall's coefficient of concordance to estimate the effect and Durbin-Conover test for post-hoc test. We performed this analysis in the R [47] package *ggstatsplot* [48]. These results are presented in Figs 2–4.

## Results

Figs 2 through 4 visually depict the influence of the centrality measures on attack extent or magnitude concerning both the number of components and the average path length in the 40 studied networks. Additionally, these figures present the statistical outcomes of the Friedman's Test and highlight significant findings obtained through the Durbin-Conover post-hoc test.

For a normal attack, the extent and magnitude of the attack on the number of components and the average path length of each centrality measure are presented in Table 1. Normal attack distributions for attack extent and magnitude of these properties can be seen in Fig 2, as well

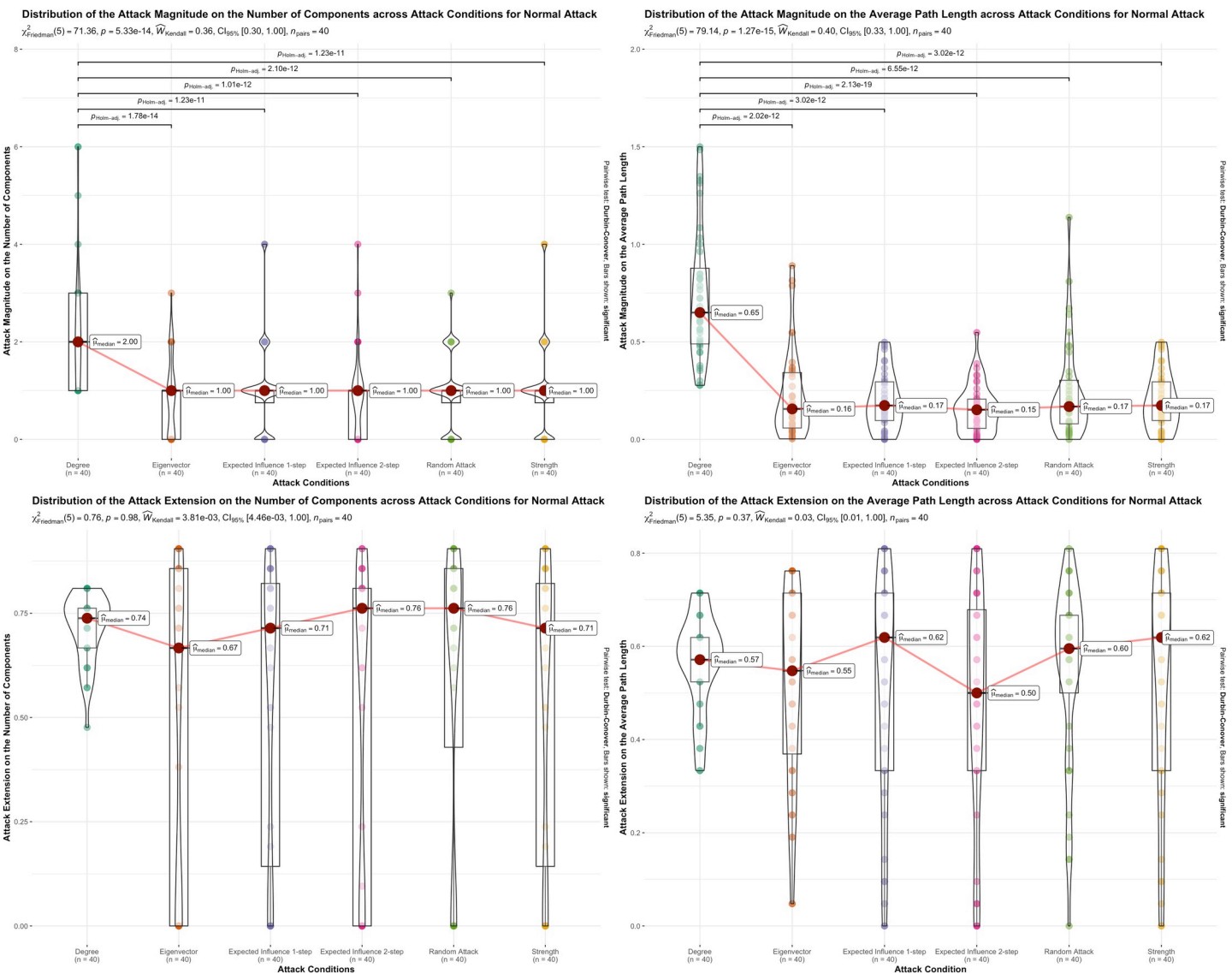

**Fig 2. Graphical representation of normal attack magnitude and extension results across the 5 attack conditions.** Graphical representation of normal attack magnitude and extension results across the 5 attack conditions, degree, eigenvector, expected influence 1-step, expected influence 2-step. In each panel result of the Friedman rank-sum test for differences between attack, and conditions are presented on top. The significant differences found between attack conditions, in the Durbin-Conover post-hoc test, are represented by lines between attack conditions and with the Holm corrected p-value above. Only significant differences are represented. Boxplots represent the interquartile range, the median and the outliers for attack magnitude or extension range for each attack condition. Violin plots display the probability density of the data.

as the significant results from the post-hoc comparison tests. We observed a statistically significant effect of centrality measures on the magnitude of the normal attack on the number of components ($X^2_F$ (5) = 71.02, p < .001, $W_k$ = 0.40, 95% CI [0.36, 0.61]) and on the average path length ($X^2_F$ (5) = 79.14, p < .001, $W_k$ = 0.43, 95% CI [0.34, 0.74]), post hoc comparisons suggest that degree centrality had a significant higher attack magnitude on the number of components and on the average path length than all the other centrality measures and the random attack. The extent of the normal attack on the number of components ($X^2_F$ (5) = 0.47, p = 0.993, $W_k$ = 0.23, 95% CI [0.20, 0.71]) and on the average path ($X^2_F$ (5) = 6.26, p = 0.281, $W_k$ = 0.33, 95% CI [0.32, 0.63]) did not show statistically significant effects.

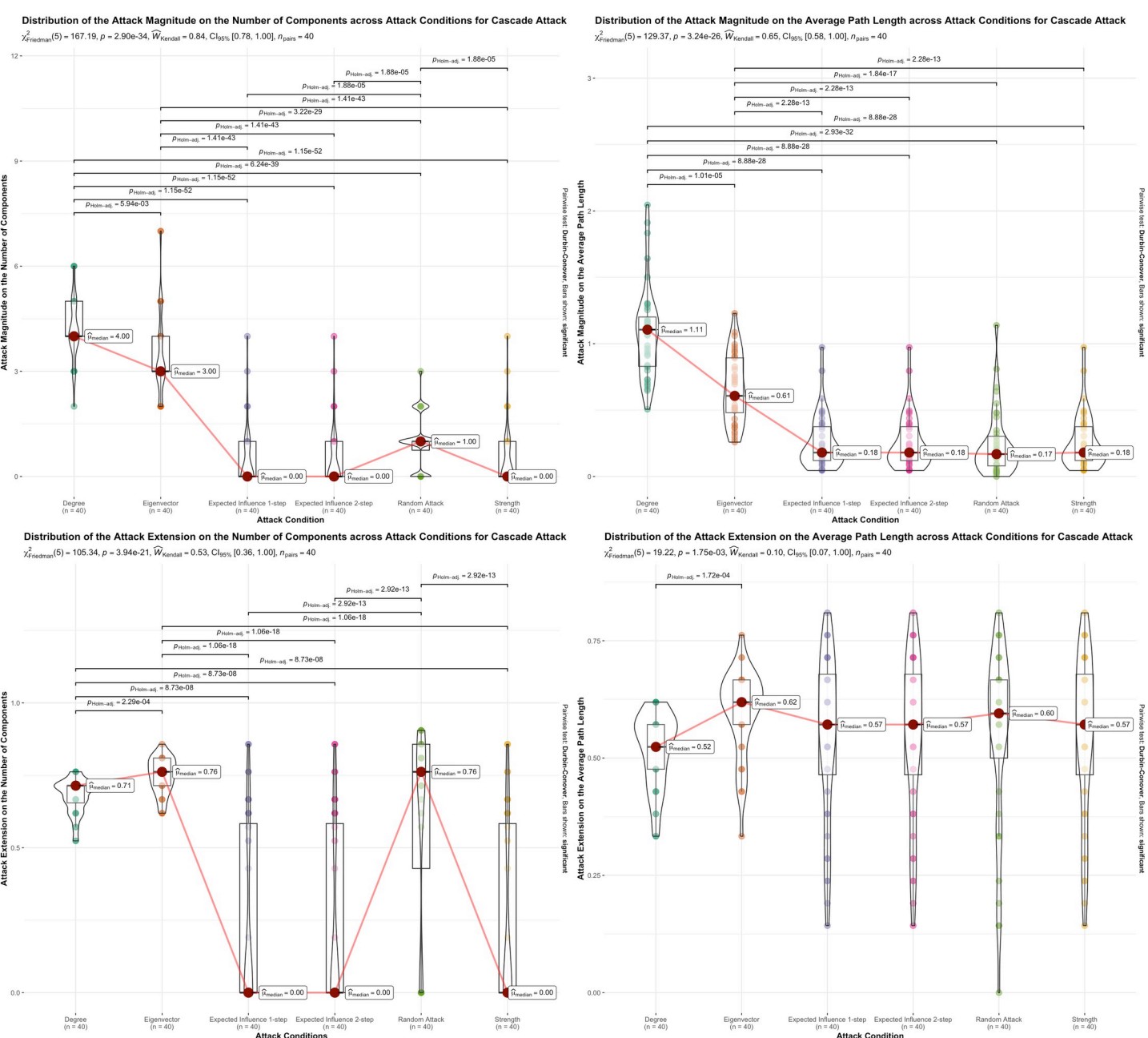

**Fig 3. Graphical representation of cascade attack magnitude and extension results across the 5 attack conditions.** Graphical representation of normal attack magnitude and extension results across the 5 attack conditions, degree, eigenvector, expected influence 1-step, expected influence 2-step. In each panel result of the Friedman rank-sum test for differences between attack, and conditions are presented. The significant differences found between attack conditions, in the Durbin-Conover post-hoc test, are represented by lines between attack conditions and with the Holm corrected p-value above. Only significant differences are represented. Boxplots represent the interquartile range, the median and the outliers for attack magnitude or extension range for each attack condition. Violin plots display the probability density of the data.

Regarding the cascade attack, the extent and magnitude of the different centrality measures are presented in Table 2, and Fig 3 shows the distributions for the different centrality measures. Magnitude of the attack showed statistically significant effect of centrality measures on the median attack magnitude on the number of components, $X^2_F (5) = 165.57$, $p < .001$, $W_k = 0.35$, 95% CI [0.33, 0.59], and on the average path $X^2_F (5) = 129.37$, $p < .001$, $W_k = 0.40$, 95%

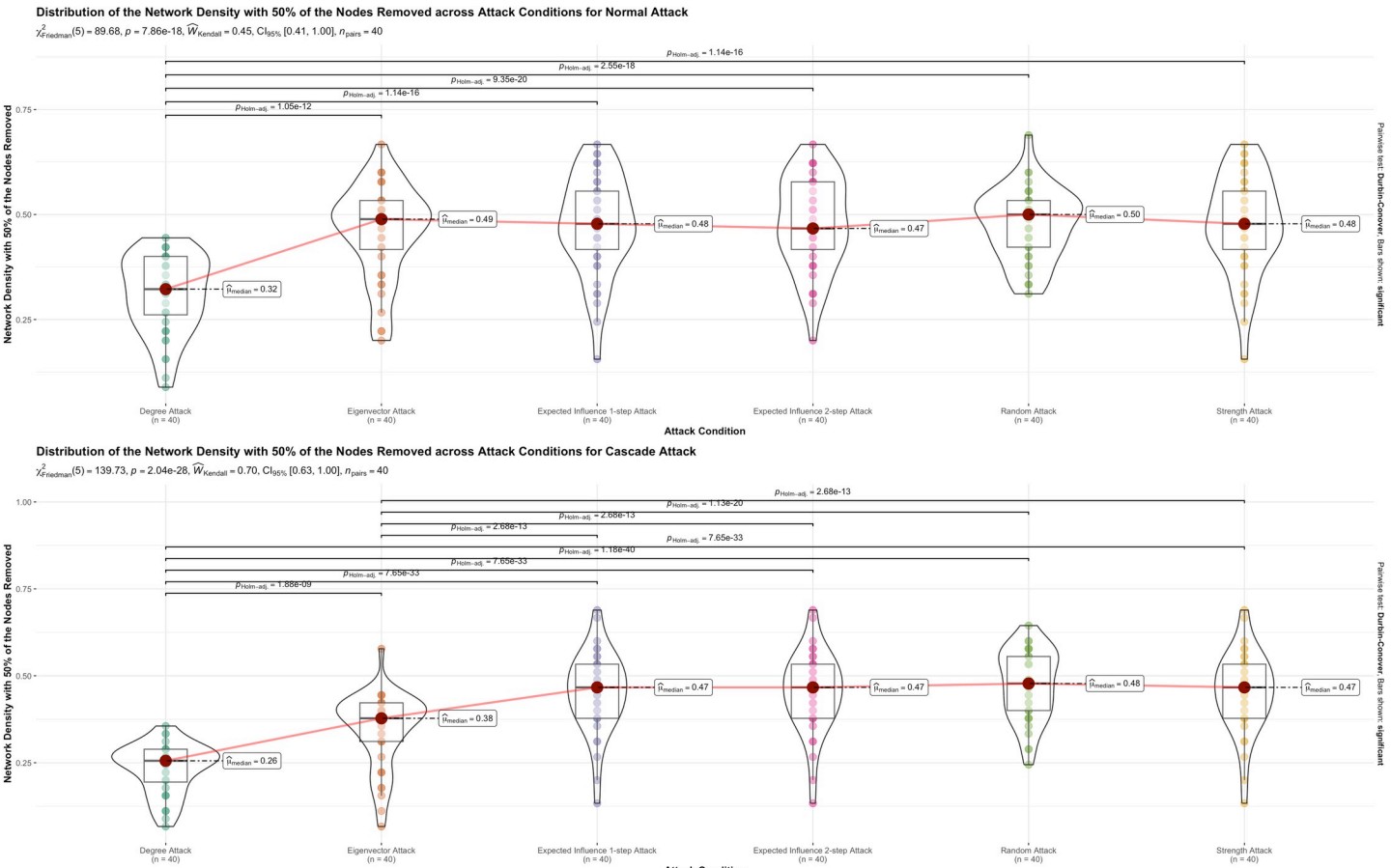

**Fig 4. Graphical representation of the results for network density with 50% of the nodes removed.** Graphical representation of the results for network density with 50% of the nodes removed in normal and cascade attack across the 5 attack conditions, degree, eigenvector, expected influence 1-step, expected influence 2-step. In each panel result of the Friedman rank-sum test for differences between attack, and conditions are presented. The significant differences found between attack conditions, in the Durbin-Conover post-hoc test, are represented by lines between attack conditions and with the Holm corrected p-value above. Only significant differences are represented. Boxplots represent the interquartile range, the median and the outliers for attack magnitude or extension range for each attack condition. Violin plots display the probability density of the data.

CI [0.32, 0.76]. Post hoc comparisons suggest that degree centrality and eigenvector centrality yielded a significantly higher attack magnitude on the number of components than the random attack and the remaining centrality measures. The random attack also had a statistically higher attack magnitude on the number of components than strength centrality and expected influence one-step and two-step. On the average path length, post hoc comparisons on the cascade attack magnitude were also significantly higher for degree centrality and eigenvector centrality. The remaining centrality measures did not present significant differences between them.

Similarly, cascade attack extent on the number of components returned a statistically significant effect $X^2_F (5) = 101.05$, $p < .001$, $W_k = 0.22$, 95% CI [0.21, 0.56] with the faster centrality measures to achieve maximum value on the number of components being expected influence one-step and two-step and strength centrality. However, as aforementioned, the attack extension is the proportion of symptoms that need to be deactivated to achieve the maximum value. Thus, the lower the score in the attack extension the better. However, expected influence one-step and two-step and strength centrality, as can be seen in the attack magnitude results, did not promote any change in the network ($M_{dn} = 0$). Accordingly, the maximum value equals

**Table 1. Descriptive statistics for normal attack.**

| Network characteristic / attack condition | Attack Magnitude | | | Attack Extent | | |
|---|---|---|---|---|---|---|
| | Mean (SD*) | Median (MAD**) | Minimum—maximum | Mean (SD) | Median (MAD) | Minimum—maximum |
| **Components** | | | | | | |
| Degree | 2.30 (1.26) | 2.00 (1.48) | 1–6 | 0.72 (0.08) | 0.74 (0.04) | 0.48–0.81 |
| Strength | 1.02 (0.92) | 1.00 (0.00) | 0–4 | 0.55 (0.36) | 0.71 (0.21) | 0–0.90 |
| Expected Influence 1-step | 1.02 (0.92) | 1.00 (0.00) | 0–4 | 0.55 (0.36) | 0.71 (0.21) | 0–0.90 |
| Expected Influence 2-step | 0.95 (0.85) | 1.00 (0.00) | 0–4 | 0.54 (0.37) | 0.76 (0.14) | 0–0.90 |
| Eigenvector | 0.88 (0.82) | 1.00 (1.48) | 0–3 | 0.49 (0.40) | 0.67 (0.35) | 0–0.90 |
| Random | 0.95 (0.85) | 1.00 (0.00) | 0–3 | 0.51 (0.36) | 0.76 (0.14) | 0–0.90 |
| **Average Path Length** | | | | | | |
| Degree | 0.74 (0.34) | 0.65 (0.28) | 0.28–1.50 | 0.56 (0.10) | 0.57 (0.07) | 0.33–0.71 |
| Strength | 0.20 (0.15) | 0.17 (0.18) | 0–0.50 | 0.52 (0.25) | 0.62 (0.21) | 0–0.81 |
| Expected Influence 1-step | 0.20 (0.15) | 0.17 (0.18) | 0–0.50 | 0.52 (0.25) | 0.62 (0.21) | 0–0.81 |
| Expected Influence 2-step | 0.16 (0.13) | 0.15 (0.12) | 0–0.55 | 0.48 (0.26) | 0.50 (0.25) | 0–0.81 |
| Eigenvector | 0.23 (0.23) | 0.16 (0.17) | 0–0.89 | 0.52 (0.21) | 0.55 (0.25) | 0.05–0.76 |
| Random | 0.28 (0.16) | 0.18 (0.18) | 0.01–0.68 | 0.59 (0.15) | 0.62 (0.14) | 0.19–0.81 |
| | 50% of nodes deactivated | | | | | |
| | Mean (SD) | Median (MAD) | | Minimum—Maximum | | |
| **Density** | | | | | | |
| Degree | 0.31 (0.09) | 0.32 (0.12) | | 0.09–0.44 | | |
| Strength | 0.47 (0.12) | 0.48 (0.12) | | 0.16–0.67 | | |
| Expected Influence 1-step | 0.47 (0.12) | 0.48 (0.12) | | 0.16–0.67 | | |
| Expected Influence 2-step | 0.48 (0.11) | 0.47 (0.13) | | 0.20–0.67 | | |
| Eigenvector | 0.46 (0.11) | 0.49 (0.08) | | 0.20–0.67 | | |
| Random | 0.47 (0.09) | 0.50 (0.08) | | 0.31–0.69 | | |

Descriptive statistics for normal attack magnitude and extent on the number of components and average path length and network density with 50% of the nodes deactivated.

* Standard deviation

** Median Absolute Deviation

the initial value of the network and, consequently, the median number of symptoms that need to be deactivated to achieve the maximum number of components is 0. The two centrality measures that promoted a change in the number of components (attack magnitude), degree and eigenvector centrality, did not show any significant differences with the random attack on the attack extent. On the average path length, the cascade attack extent presented a statistically significant difference, $X^2_F (5) = 19.89$, p = 0.001, $W_k = 0.50$, 95% CI [0.45, 0.89] with post hoc comparisons suggesting a better performance of degree centrality over the other centrality measures and random attack.

Concerning network density with 50% of the symptoms deactivated for normal and cascade attacks, distributions can be found in Fig 4. Normal attack showed significant effects between centrality measures, $X^2_F (5) = 89.68$, p < .001, $W_k = 0.70$, 95% CI [0.66, 0.94]. Post hoc comparisons suggest that degree centrality has a significantly higher impact in reducing the network density in comparison to all other centrality measures and the random attack. For the cascade attack significant effects were also found, $X^2_F (5) = 139.08$, p < .001, $W_k = 0.69$, 95% CI [0.67, 0.88]. As observed in the post hoc comparisons, the eigenvector centrality cascade attack promoted a higher decrease in network density than all other centrality measures, except degree centrality, and the random attack. Degree centrality promoted the most

**Table 2. Descriptive statistics for cascade attack.**

| Network characteristic / attack condition | Attack Magnitude | | | Attack Extent | | |
|---|---|---|---|---|---|---|
| | Mean (SD*) | Median (MAD**) | Minimum—maximum | Mean (SD) | Median (MAD) | Minimum—maximum |
| **Components** | | | | | | |
| Degree | 4.20 (0.88) | 4.00 (0.74) | 2.00–6.00 | 0.68 (0.06) | 0.71 (0.07) | 0.52–0.76 |
| Strength | 0.55 (0.90) | 0.00 (0.00) | 0.00–4.00 | 0.23 (0.32) | 0.00 (0.00) | 0.00–0.86 |
| Expected Influence 1-step | 0.55 (0.90) | 0.00 (0.00) | 0.00–4.00 | 0.23 (0.32) | 0.00 (0.00) | 0.00–0.86 |
| Expected Influence 2-step | 0.55 (0.90) | 0.00 (0.00) | 0.00–4.00 | 0.23 (0.32) | 0.00 (0.00) | 0.00–0.86 |
| Eigenvector | 3.50 (1.24) | 3.00 (1.48) | 2.00–7.00 | 0.75 (0.06) | 0.76 (0.07) | 0.62–0.86 |
| Random | 0.98 (0.73) | 1.00 (0.00) | 0.00–3.00 | 0.59 (0.36) | 0.76 (0.14) | 0–0.90 |
| **Average Path Length** | | | | | | |
| Degree | 1.09 (0.35) | 1.11 (0.28) | 0.51–2.05 | 0.52 (0.08) | 0.52 (0.07) | 0.33–0.62 |
| Strength | 0.26 (0.21) | 0.18 (0.13) | 0.05–0.97 | 0.55 (0.17) | 0.57 (0.18) | 0.14–0.81 |
| Expected Influence 1-step | 0.26 (0.21) | 0.18 (0.13) | 0.05–0.97 | 0.55 (0.17) | 0.57 (0.18) | 0.14–0.81 |
| Expected Influence 2-step | 0.26 (0.21) | 0.18 (0.13) | 0.05–0.97 | 0.55 (0.17) | 0.57 (0.18) | 0.14–0.81 |
| Eigenvector | 0.66 (0.26) | 0.61 (0.32) | 0.26–1.23 | 0.60 (0.09) | 0.62 (0.07) | 0.33–0.76 |
| Random | 0.25 (0.24) | 0.18 (0.18) | 0.00–1.14 | 0.55 (0.19) | 0.62 (0.14) | 0.00–0.81 |
| | 50% of nodes deactivated | | | | | |
| | Mean (SD) | Median (MAD) | Minimum—Maximum | | | |
| **Density** | | | | | | |
| Degree | 0.23 (0.07) | 0.26 (0.05) | 0.07–0.36 | | | |
| Strength | 0.45 (0.11) | 0.47 (0.10) | 0.13–0.69 | | | |
| Expected Influence 1-step | 0.45 (0.11) | 0.47 (0.10) | 0.13–0.69 | | | |
| Expected Influence 2-step | 0.45 (0.11) | 0.47 (0.10) | 0.13–0.69 | | | |
| Eigenvector | 0.35 (0.11) | 0.38 (0.07) | 0.07–0.58 | | | |
| Random | 0.47 (0.10) | 0.48 (0.12) | 0.24–0.64 | | | |

Descriptive statistics for cascade attack magnitude and extent on the number of components and average path length and network density with 50% of the nodes deactivated.

* Standard deviation

** Median Absolute Deviation

significant reduction of the network density, outperforming all other measures, including eigenvector centrality, and random attack.

Attending to these results, a post hoc analysis was made to compare the magnitude and extent of the effect of degree centrality under normal and cascade attacks on the network characteristics. Results suggest that the degree centrality cascade attack outperformed the normal attack in all the network properties examined, yielding a higher magnitude, a lower extension, and density. Results from this analysis are presented in the S1 Appendix.

## Discussion

Our findings indicate that the most significant alterations in the network properties primarily manifest through degree-based attacks. Notably a degree-based normal attack also exerts a more substantial influence in the network properties than any other centrality measure studied, a result consistent with prior study [27]. Furthermore, in the context of cascade attacks, eigenvector centrality emerges as the central measure with the greatest impact on the network, surpassing all other centrality measures except for degree. This observation aligns with earlier research by Dablander and colleagues [19], which suggested that eigenvector centrality serves as a superior proxy for causality compared to the commonly employed centrality measures in

psychological networks. These findings carry significant implications for the field of psychopathology within the framework of network theory.

Network theory of psychopathology has suggested that central symptoms might be valuable therapeutic targets, due to their proposed ability to fasten the deactivation of connections between symptoms [8, 13–16, 25, 49–51]. This is one of the core propellers of the network theory of psychopathology, that lead to its growth in recent years [52, 53]. However, evidence for this hypothesis is still scarce with studies focusing on cross-sectional networks and grounding the identification of possible therapeutic targets on the initial estimations of centrality measures [21–23, 27]. Due to these inconclusive results, it has been recognized that there are changes in symptoms centrality that occur during treatment [54] and that idiographic networks might be more appropriate to identify treatment targets [35]. This might have important implications for treatment personalization.

In this context, we explored the impact of deactivating symptoms in contemporaneous idiographic networks through two distinct procedures. The first is based on a single time point estimate of network centrality (normal attack), and a second procedure, where, after each symptom deactivation, centrality measures are estimated again (cascade attack). The impact of symptom deactivation was assessed through a set of network properties since it has been suggested changes in the network density might be able to differentiate between different clinical presentations [36, 55]. However, due to the conflicting results in previous studies regarding the association between symptoms' remission and networks connectivity [20, 56] and the identified changes in the network topology [28–33] we have explored the impact of symptom deactivation in two more network properties, average path length and the number of components.

Globally, our results suggest that changes in psychopathological network structure are best achieved through degree centrality. In comparison with the most common centrality metrics in psychopathological networks (i.e., strength centrality and expected influence one-step and two-step), the deactivation of symptoms by the absolute number of connections (i.e., degree centrality) seems to have a higher impact on the network structure. A previous study using cross-sectional networks [27] also found that degree centrality was the only centrality measure that was able to produce significant changes in the network structure. However, this study [27] only found significant changes in the number of components of the networks. In turn, the present study suggests that for contemporaneous within-person networks all the three network properties examined are transformed through a degree-based attack. These results suggest that different properties might be of interest according to the nature of the network (nomothetic or idiographic). In fact, previous research has also pointed to this need of further exploration and clarification of the properties of interest in psychological network [57] and the impact of these networks' structural properties in the selection of centrality measures [58].

With the field focusing on the strength centrality and expected influence measures to identify important symptoms in the network, it's of relevance that neither of these measures was able to promote significant changes in the network structure. Interestingly, the random deactivation of symptoms revealed a significantly higher impact magnitude in the number of components than a cascade attack through expected influence one and two-step and strength centrality. Thus, if changes in a person's symptomatology are identifiable by changes in the network structure, the traditional psychopathological centrality metrics do not seem able to induce significant changes. Consequently, this might explain the inconclusive results in previous studies that explored if these centrality measures were related to changes in symptomatology [21–23].

It has been suggested that all centrality measures make implicit assumptions about the network processes of node-to-node transmission and the type of trajectories followed [58, 59]. The case may be that common centrality metrics in psychopathological networks are not accessing the specific processes that occur in these networks or are accessing some other

processes that are not related to network transformation. For example, they might be identifying emergent phenomena in the network that need to be addressed (e.g., a very active symptom) but not phenomena related to disorder maintenance (e.g., symptoms that sustain the disorder). However, in psychological networks, the processes within the networks that generate and maintain mental disorders are still unknown. Interestingly, with a cascade attack, eigenvector centrality produced significant changes in the network structure. This might be due to its suggested proximity to the causality structure of the network [19] and might mean that this measure is tapping into a specific process in psychological networks. Understanding these processes will advance the identification of treatment targets by enabling an enhanced selection of centrality metrics.

Besides exploring which network centrality measure promoted changes in the network structure, we have also tested two types of attacks, normal and cascade. Although degree centrality had a better performance than any other measure in both attacks. In the cascade attack, the magnitude and the extension were significantly higher than in a normal attack. This suggests that it might be of importance to estimate symptoms' centrality each time before an intervention is deployed to act on the symptom with the highest degree at any given time point. The dynamic fluctuations of central symptoms during a psychotherapeutic process have been highlighted by previous studies [54] and our results suggest that assessing and intervening in which symptom is central at any given time-point might produce faster recoveries. Consequently, the estimation at a single time-point of centrality measures to establish treatment targets for intervention might not be the most effective procedure to promote changes in the network structure.

This has important implications for treatment personalization. Our results suggest that to promote more effective treatments the assessment of the central symptoms must be done each time before the intervention is done. Meaning that, in the context of idiographic networks, symptomatology needs to continuously be assessed through, for example, ecological momentary assessments [60, 61] for the duration of treatment to determine, at each session, in which symptom the treatment should focus. This has not been the current practice on open trial studies using centrality metrics for treatment target identification and intervention guidance [37, 62]. This, in addition to the positive results that these studies have shown, raises an important question about the specificity of the psychotherapeutic strategies. Do psychotherapeutic strategies and psychopharmacological treatments have the specificity needed to act on a specific central symptom at each time? And is this a negative constraint of the treatments or is it a positive consequence? The answers to these questions are still unknown but the first results seem promising [63, 64]. Although targeting psychological symptoms and behaviors is inherently distinct from targeting genes or computer networks, recent studies [63–68] have started to reveal contrasting effects not only among different treatment modalities [65–68], but also throughout the course of treatment [63, 64]. These early results suggest that achieving a remarkable level of precision in psychological treatments may be attainable. This in addition with the selection of treatment targets through network centrality measures has the potential to position psychology on the forefront of precision medicine [69–71] with more effective, precise, and personalized treatment strategies.

Besides this, some limitations of our study should be pointed out. First, both of our procedures remove the deactivated symptoms from the network. This might be a strong assumption for some of the symptoms. For example, anxiety might always fluctuate at lower levels, without ever being completely absent of the network. In turn, insomnia or obsessions and compulsions might in fact be completely absent of the network if a person does not have a psychopathology. In our study we treated all symptoms equally, assuming that all symptoms would be absent from the network after treatment. However, this is in fact a strong assumption and future

studies should explore this question. Moreover, our study only targets nodes and some interventions might not act on the nodes its selves but rearrange the connections between them. With this in mind, we think future studies with frameworks like the one proposed by Blanken and colleagues [64] might provide important insights about the specificity of psychological interventions.

Secondly, we used only centrality measures, leaving another important concept of psychopathological networks, bridge symptoms, outside of our study. Bridge symptoms have been proposed as symptoms that connect two different disorders and that acting in these symptoms might promote a faster disintegration of a comorbidity network [72, 73]. Our network is a comorbidity network comprising symptoms of depression and anxiety and identifying and deactivating bridge symptoms might had led to faster deactivation of the network. Furthermore, there are several unexplored centrality measures that can be explored in the context of psychopathological networks, such as the recently developed hybrid centrality measure [74, 75]. Hybrid centrality measures group together several rankings of other centrality measures [75] and can potentially yield valuable insights for the refinement of treatment target selection.

In addition, since we have used within-person contemporaneous networks our centrality measures lack the directionality that could be obtained with the use of temporal networks. Nevertheless, due to the strong model assumptions about the temporal effects [38], meaning that all relevant temporal symptom dynamics can be captured in the ESM or EMA time scale, we opted for contemporaneous networks. However, with passive data [76–80], future studies can surpass the temporal dynamics problem that emerge from ESM and EMA methods since passive data can be collect continuously.

Another important limitation in our work is that we assume, as has been proposed in previous studies [36, 55], that network properties identify psychological states, although this hypothesis is currently lacking consistent evidence [20, 56]. In previous studies that focused on the idiographic network, a relationship between network density and psychopathological states was found [36, 55] and our results show that there are clear changes in the network density after the deactivation of 50% of the symptoms. However, we also use two network properties rarely studied in psychopathological networks and without a clear theoretical formulation, although we think it's important to explore these new properties, we also acknowledge that there's a need to theoretically frame these properties.

Finally, a sensitivity analysis based on demographic variables such as sex or age could provide further insights into the selection of treatment targets, by understanding if different centrality measures are better suited for different groups of the population.

## Conclusion

Our study provides the first simulation study in idiographic networks to examine symptom deactivation through several centrality measures. The emergence of degree centrality as a measure more suitable to transform the network might be of relevance for further studies trying to identify treatment targets through network analysis. Further exploration of network properties is needed, but, if changes in the network structure are aligned with psychopathological and healthy states, deactivating symptoms through a cascade attack based on degree centrality might promote faster and more effective treatments.

## Supporting information

**S1 Appendix. Participants characteristics.** Network Basic Features. Graphical representation of the comparison between a degree-based normal attack and a degree-based cascade attack for the number of components and the average path length. Graphical representation of the

comparison between a degree-based normal attack and a degree-based cascade attack for network density with 50% of the nodes removed. Plots Displaying Attack Results for Each Individual Network.
(DOCX)

## Author Contributions

**Conceptualization:** Daniel Castro, Tiago Bento Ferreira.

**Data curation:** Daniel Castro.

**Formal analysis:** Daniel Castro.

**Investigation:** Daniel Castro.

**Methodology:** Daniel Castro.

**Project administration:** Tiago Bento Ferreira.

**Resources:** Daniel Castro.

**Software:** Daniel Castro.

**Supervision:** Fernando Ferreira-Santos, Tiago Bento Ferreira.

**Writing – original draft:** Daniel Castro.

**Writing – review & editing:** Daniel Castro, Deisy Gysi, Filipa Ferreira, Tiago Bento Ferreira.

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
