## [Decision Letter · Decision Letter 0]

7 Sep 2023

PONE-D-23-24252Centrality Measures in Psychological Networks: A Simulation Study on Identifying Effective Treatment TargetsPLOS ONE

Dear Dr Daniel Castro,

Thank you for submitting your manuscript to PLOS ONE. After careful consideration, we feel that it has merit but does not fully meet PLOS ONE’s publication criteria as it currently stands. Therefore, we invite you to submit a revised version of the manuscript that addresses the points raised during the review process.

We look forward to receiving your revised manuscript.

Kind regards,

Mu-Hong Chen, M.D., Ph.D.

Academic Editor

PLOS ONE

Journal Requirements:

This work was supported by national funding from the Portuguese Foundation for Science and Technology (UIDB/00050/2020). The first author is supported by the Portuguese Foundation for Science and Technology through the Ph.D. grant: SFRH/BD/148884/2019.

3. We noted in your submission details that a portion of your manuscript may have been presented or published elsewhere.Please clarify whether this [conference proceeding or publication] was peer-reviewed and formally published. If this work was previously peer-reviewed and published, in the cover letter please provide the reason that this work does not constitute dual publication and should be included in the current manuscript.

Editor and Reviewer comments: 

Reviewer 1

Network analysis has long been a tool for assessing psychological disorders, yet opinions on its value and therapeutic implications remain divided. Given this, the study's focus on dynamic changes in networks through contemporaneous idiographic networks is commendable.

I concur with the authors that idiographic networks offer a more targeted approach to understanding symptoms. However, considering each patient possesses a unique symptom network, the study's collective network analysis might not be as conclusive when applied individually in treatment. Was the in-person network considered in the study? The authors' insights on this would be valuable.

It would enhance comprehension if basic network features, possibly presented in Table 1, were visualized in a figure, offering readers a clearer understanding of the studied symptom networks.

The term "symptom deactivation" needs elucidation. Is it indicative of symptom improvement during treatment or does it highlight symptoms that naturally waned without intervention?

While the study's strength lies in its serial follow-up of the network, the limited sample size poses concerns. A presentation of a network stability coefficient might aid in ensuring reliable interpretation.

The differentiation between normal and cascade-based attacks could benefit from a tangible example. As it stands, the exposition seems tailored for those deeply familiar with network statistical analysis. Simplifying the explanation might cater to a wider audience.

The figures would be more impactful with a detailed textual interpretation.

Although the primary goal of the study is stated as contrasting the two approaches, the organization and interpretation of the results don't align with this. Structuring the content based on the study's declared intent could enhance clarity.

Lastly, the Discussion section would benefit from a tighter organization. A succinct summary of the primary findings at the outset, followed by an interpretative comparison with previous studies rather than a mere summary, would give the discussion more depth and relevance.

Reviewer 2

This study utilizes a dataset accessible at [https://osf.io/mgdp6] and involves a sample of 40 participants, comprising 25 individuals with generalized anxiety disorder and 15 with major depressive disorder. The authors conducted a nuanced analysis of contemporaneous idiographic networks that characterize depression and anxiety symptoms within this cohort. Comparisons were made between cascade-based attacks and normal attacks. The authors suggest that degree centrality, as a more effective measure of network transformation, has considerable implications for future research aimed at delineating treatment goals through network analysis. Current evidence suggests that targeting symptoms through degree centrality-based cascading approaches may expedite more effective therapeutic interventions if changes in network structure correspond to psychopathology and health status.

The paper is overall well written and the message is clear. I have some minor comments as follows.

Overall

The author needs to define the first occurrence of the abbreviation. For example, "SD" in Method, Page 6; “LASSO” in Method, Page 7; “M” and “MAE” in Table.

Introduction, Page 4

What is “moderate support”? The description of “moderate” is a bit vague and difficult to understand.

Method

Did the authors consider the effects of sex and age differences in their study? Conducting sensitivity analyzes based on demographic variables such as sex or age can provide additional insights.If these factors were not included in the analysis, this may be a limitation of the study.

Table

The number of digits after the decimal point should remain consistent throughout the manuscript.

Figure

The figure legend appears to be incomplete, such as "degree, eigenvector, expected influence 1-step, and expected influence 2-step."

Reviewer 3

The reviewed paper is well planned, precise, technically elegant, cited sources are adequate, fresh, sound and well searched. The minor problems however should be solved as indicated below:

1. R package psychNetsAttack is not easy (or impossible?) to access and install (as not so much experienced user I cannot clearly distunguish the latter two; I tried to update my R and RStudio and tried again, unsuccessfully again. BTW I was able to install and implement other (several) R packages). My suggestion is to try (the Editorial Staff or Authors asked by Editors) to load and use the dedicated R package because it is crucial for- and for many readers also important benefit from the reviewed paper. If it works on training dataset then all is OK (so I regret I had no chance to install and launch it on my workstation and my datasets). If installation and/or package is not working, that should be amended before paper is published. Or, the phrases suggesting that every researcher can use the package and repeat reasoning on his/her datasets, should be corrected.

2. The so called 'slim finger' versus 'fat finger' problem is to be discussed in context of influencing nodes in the psychopathological networks in contrast to more technical problems (e.g. net of PCs) or biology (switching off the genes). It is a bit touched with the phrase (322-323): "Do psychotherapeutic strategies and psychopharmacological treatments have the specificity needed to act on a specific central symptom at each time?"

3. Freshly invented centrality measures like hybrid centrality may be mentioned. Same is true for bridge centrality (bridges are discussed).

Regards and Good Luck!

Reviewers' comments:

Reviewer's Responses to Questions

**Comments to the Author**

1. Is the manuscript technically sound, and do the data support the conclusions?

Reviewer #1: Yes

Reviewer #2: Yes

Reviewer #3: Yes

2. Has the statistical analysis been performed appropriately and rigorously? 

Reviewer #1: Yes

Reviewer #2: I Don't Know

Reviewer #3: Yes

3. Have the authors made all data underlying the findings in their manuscript fully available?

Reviewer #1: Yes

Reviewer #2: Yes

Reviewer #3: Yes

4. Is the manuscript presented in an intelligible fashion and written in standard English?

Reviewer #1: Yes

Reviewer #2: Yes

Reviewer #3: Yes

5. Review Comments to the Author

Reviewer #1: Network analysis has long been a tool for assessing psychological disorders, yet opinions on its value and therapeutic implications remain divided. Given this, the study's focus on dynamic changes in networks through contemporaneous idiographic networks is commendable.

I concur with the authors that idiographic networks offer a more targeted approach to understanding symptoms. However, considering each patient possesses a unique symptom network, the study's collective network analysis might not be as conclusive when applied individually in treatment. Was the in-person network considered in the study? The authors' insights on this would be valuable.

It would enhance comprehension if basic network features, possibly presented in Table 1, were visualized in a figure, offering readers a clearer understanding of the studied symptom networks.

The term "symptom deactivation" needs elucidation. Is it indicative of symptom improvement during treatment or does it highlight symptoms that naturally waned without intervention?

While the study's strength lies in its serial follow-up of the network, the limited sample size poses concerns. A presentation of a network stability coefficient might aid in ensuring reliable interpretation.

The differentiation between normal and cascade-based attacks could benefit from a tangible example. As it stands, the exposition seems tailored for those deeply familiar with network statistical analysis. Simplifying the explanation might cater to a wider audience.

The figures would be more impactful with a detailed textual interpretation.

Although the primary goal of the study is stated as contrasting the two approaches, the organization and interpretation of the results don't align with this. Structuring the content based on the study's declared intent could enhance clarity.

Lastly, the Discussion section would benefit from a tighter organization. A succinct summary of the primary findings at the outset, followed by an interpretative comparison with previous studies rather than a mere summary, would give the discussion more depth and relevance.

Reviewer #2: This study utilizes a dataset accessible at [https://osf.io/mgdp6] and involves a sample of 40 participants, comprising 25 individuals with generalized anxiety disorder and 15 with major depressive disorder. The authors conducted a nuanced analysis of contemporaneous idiographic networks that characterize depression and anxiety symptoms within this cohort. Comparisons were made between cascade-based attacks and normal attacks. The authors suggest that degree centrality, as a more effective measure of network transformation, has considerable implications for future research aimed at delineating treatment goals through network analysis. Current evidence suggests that targeting symptoms through degree centrality-based cascading approaches may expedite more effective therapeutic interventions if changes in network structure correspond to psychopathology and health status.

The paper is overall well written and the message is clear. I have some minor comments as follows.

Overall

The author needs to define the first occurrence of the abbreviation. For example, "SD" in Method, Page 6; “LASSO” in Method, Page 7; “M” and “MAE” in Table.

Introduction, Page 4

What is “moderate support”? The description of “moderate” is a bit vague and difficult to understand.

Method

Did the authors consider the effects of sex and age differences in their study? Conducting sensitivity analyzes based on demographic variables such as sex or age can provide additional insights.If these factors were not included in the analysis, this may be a limitation of the study.

Table

The number of digits after the decimal point should remain consistent throughout the manuscript.

Figure

The figure legend appears to be incomplete, such as "degree, eigenvector, expected influence 1-step, and expected influence 2-step."

Reviewer #3: The reviewed paper is well planned, precise, technically elegant, cited sources are adequate, fresh, sound and well searched. The minor problems however should be solved as indicated below:

1. R package psychNetsAttack is not easy (or impossible?) to access and install (as not so much experienced user I cannot clearly distunguish the latter two; I tried to update my R and RStudio and tried again, unsuccessfully again. BTW I was able to install and implement other (several) R packages). My suggestion is to try (the Editorial Staff or Authors asked by Editors) to load and use the dedicated R package because it is crucial for- and for many readers also important benefit from the reviewed paper. If it works on training dataset then all is OK (so I regret I had no chance to install and launch it on my workstation and my datasets). If installation and/or package is not working, that should be amended before paper is published. Or, the phrases suggesting that every researcher can use the package and repeat reasoning on his/her datasets, should be corrected.

2. The so called 'slim finger' versus 'fat finger' problem is to be discussed in context of influencing nodes in the psychopathological networks in contrast to more technical problems (e.g. net of PCs) or biology (switching off the genes). It is a bit touched with the phrase (322-323): "Do psychotherapeutic strategies and psychopharmacological treatments have the specificity needed to act on a specific central symptom at each time?"

3. Freshly invented centrality measures like hybrid centrality may be mentioned. Same is true for bridge centrality (bridges are discussed).

Regards and Good Luck!

6. PLOS authors have the option to publish the peer review history of their article (what does this mean?). If published, this will include your full peer review and any attached files.

Reviewer #1: **Yes: **Hong Jun Jeon

Reviewer #2: No

Reviewer #3: No

---

## [Author Response · Author response to Decision Letter 0]

4 Dec 2023

Dear Reviewers,

Thank you for taking the time to review our manuscript “Centrality Measures in Psychological Networks: A Simulation Study on Identifying Effective Treatment Targets”. We appreciate your valuable feedback and suggestions, which have helped us improve the quality of our work. We have carefully considered your comments and believe we made the necessary revisions to address your concerns.

Reviewer 1

I concur with the authors that idiographic networks offer a more targeted approach to understanding symptoms. However, considering each patient possesses a unique symptom network, the study's collective network analysis might not be as conclusive when applied individually in treatment. Was the in-person network considered in the study? The authors' insights on this would be valuable.

- We agree with the reviewer that individual pattern of evolution of each participant is important and, due to that, in the Supplementary Materials the changes caused by our simulation to each in-person network can be seen. We used a contemporary network of each participant, and targeted symptoms based on the centrality measures for that participant in specific. In the manuscript we aggregated the results from each participant to understand the overall effect of selecting treatment targets through centrality measures. We have added this information to the manuscript.

p.8, line 196 - 205

To assess the impact of symptom deactivation on the network density we computed the density of the network at 50% of the symptoms deactivated. All these analyses were performed in the package psychNetsAttack [48] for R [49]. We did this to each of the 40 networks (Fig. S3 – S83 in the Supplementary Materials) and then aggregated the results from each of the 40 networks to compare the results of each centrality measure.

To compare each centrality measure attack magnitude and extent on the number of components, average path length, and network density for both normal and cascade attack we used a Friedmann’s Test, we followed by a Kendall’s coefficient of concordance to estimate the effect and Durbin-Conover test for post-hoc test. We performed this analysis in the R [49] package ggstatsplot [50]. These results are presented in Figures 2 – 4. 

The term "symptom deactivation" needs elucidation. Is it indicative of symptom improvement during treatment or does it highlight symptoms that naturally waned without intervention?

- In our study symptom deactivation is a proxy for symptom improvement, in the sense that the symptom that is deactivated is removed from the network in order to remove its connections with the other symptoms. This is done to simulate the effect of detecting a central symptom and then acting on it and improving it until the symptom does not exist. To respond to the reviewer comment we have clarified this in the manuscript.

p.7, line 162 - 166

We investigate symptom deactivation as an indirect measure of symptom improvement, operating under the assumption that a symptom’s complete recovery would result in its removal from the network of symptoms. This is done to simulate the effect of detecting a central symptom and then acting on it and improving it until the symptom is not felt / reported by participants. 

While the study's strength lies in its serial follow-up of the network, the limited sample size poses concerns. A presentation of a network stability coefficient might aid in ensuring reliable interpretation.

- We agree with the reviewer that the network stability it’s important for a reliable interpretation. However, we would like to highlight that for each of the 40 idiographic networks analyzed to be constructed there were a mean of 130 observations and that the networks used were also previously published (Fischer et al., 2017). On their original study model stability was assessed through RMSEA, Browns chi-square goodness-of-fit test, and the CFI. We have now added this information to the manuscript, including the criteria for stability determined by the original authors as well as an overview of the results. To ensure reproducibility of our study we have also made all the materials open, including the R package constructed for this study on https://osf.io/k2z84/. We have now addressed this very important comment on our manuscript.

p.6 -7, line 154 - 159

On the original study [6], model fit was assessed with RMSEA, Browns chi-square goodness-of-fit test, and the CFI. The authors considered non-significant chi-square tests, RMSEA values less than .060 and CFI values equal or greater than .95 to reflect an excellent fit. All participants exhibited an excellent fit on both chi-square and CFI. On RMSEA only one participant had a value of 0.062, with all the other participants exhibiting values below 0.060.

Fisher AJ, Reeves JW, Lawyer G, Medaglia JD, Rubel JA. Exploring the idiographic dynamics of mood and anxiety via network analysis. J Abnorm Psychol. 2017;126: 1044–1056. doi:10.1037/abn0000311 

It would enhance comprehension if basic network features, possibly presented in Table 1, were visualized in a figure, offering readers a clearer understanding of the studied symptom networks.

The differentiation between normal and cascade-based attacks could benefit from a tangible example. As it stands, the exposition seems tailored for those deeply familiar with network statistical analysis. Simplifying the explanation might cater to a wider audience.

- To address these two comments from the reviewer, we have incorporated a figure into the manuscript. This figure aims to enhance the understanding of the two types of attacks and provides an illustrative example of the network's features evolution throughout the process. We also changed the explanation of each attack to cater to a wider audience as requested by the reviewer.

p. 7, line 167 - 177

The network exploration of symptom deactivation is performed as a cascade-based attack and a normal attack. In a cascade-based attack, symptoms are deactivated in their decreasing order according to their centrality, which is iteratively calculated at every symptom removal. In normal attack [28], symptoms were deactivated according to their original centrality. We compare this with random attack symptom deactivation procedure, where symptoms are randomly deactivated. In each type of attack a symptom is identified as a treatment target and deactivated from the network. For the cascade-based attack, symptom networks and treatment target selection are constantly being estimated and selected after each symptom deactivation. In the normal attack the treatment targets order is determined based on the initial symptom network of the participant and the attack follows that order without estimating the network again. Figure 1 illustrates an example of each attack and their differential impact on the average path length.

The figures would be more impactful with a detailed textual interpretation.

- We added to the results section a detailed explanation of the figures and added more information to the figure’s legends.

p. 9, line 208 - 211

Figures 2 through 4 visually depict the influence of the centrality measures on attack extent or magnitude concerning both the number of components and the average path length in the 40 studied networks. Additionally, these figures present the statistical outcomes of the Friedman’s Test and highlight significant findings obtained through the Durbin-Conover post-hoc test. 

Although the primary goal of the study is stated as contrasting the two approaches, the organization and interpretation of the results don't align with this. Structuring the content based on the study's declared intent could enhance clarity.

- Our primary goal, as stated in the introduction, is to test the theoretical proposal that central symptoms are valuable therapeutic targets. To test this hypothesis, we compare both approaches, cascade and normal attack to random deactivation of symptoms. To address the reviewer comment we have clarified this in the manuscript. We also believe that the way the data analyses was explained in the previous version could leave to the assumption that our main objective was to compare both attacks. With the previous responses to the reviewer this was addressed to. 

p. 5 line 111 - 117

Considering this, the hypothesis that central symptoms might be valuable therapeutic targets remains open. Here we address this hypothesis by assessing the impact of symptom deactivation on idiographic networks according to different centrality measures. We do it, by comparing the impact of symptom deactivation using two different types of symptom deactivation procedures, one procedure based only on a single point estimate of centrality measures and a second procedure in which the centrality measures are re-estimated every time a symptom is deactivated. As usual in network science, the impact of symptom deactivation according to the centrality estimates is evaluated in comparison to the random deactivation of symptoms.

Lastly, the Discussion section would benefit from a tighter organization. A succinct summary of the primary findings at the outset, followed by an interpretative comparison with previous studies rather than a mere summary, would give the discussion more depth and relevance.

- To comply with the reviewer comment we have adjusted the discussion section.

p. 11, line 267 – 276

Our findings indicate that the most significant alterations in the network properties primarily manifest through degree-based attacks. Notably a degree-based normal attack also exerts a more substantial influence in the network properties than any other centrality measure studied, a result consistent with prior study [28]. Furthermore, in the context of cascade attacks, eigenvector centrality emerges as the central measure with the greatest impact on the network, surpassing all other centrality measures except for degree. This observation aligns with earlier research by Dablander and colleagues [20], which suggested that eigenvector centrality serves as a superior proxy for causality compared to the commonly employed centrality measures in psychological networks. These findings carry significant implications for the field of psychopathology within the framework of network theory.

Reviewer 2

The author needs to define the first occurrence of the abbreviation. For example, "SD" in Method, Page 6; “LASSO” in Method, Page 7; “M” and “MAE” in Table.

- We agree with the reviewer and define the first occurrence of each abbreviation.

What is “moderate support”? The description of “moderate” is a bit vague and difficult to understand (p.4)

- We agree with the reviewer and changed the sentence.

p. 4 line 82 - 84

The authors concluded that symptom centrality was not generalized across measures and frequency of symptom endorsement also predicted change while being generalized across measures.

Did the authors consider the effects of sex and age differences in their study? Conducting sensitivity analyzes based on demographic variables such as sex or age can provide additional insights.If these factors were not included in the analysis, this may be a limitation of the study 

- We agree with the reviewer that sensitivity analysis based on demographic variables such as sex and age might provide important insights. As we did not conduct this type of analysis, we added this to the limitations of our study in the discussion section. 

p. 15 line 380 - 382

Finally, a sensitivity analysis based on demographic variables such as sex or age could provide further insights into the selection of treatment targets, by understanding if different centrality measures are better suited for different groups of the population. 

The number of digits after the decimal point should remain consistent throughout the manuscript.

- We have corrected this throughout the manuscript.

The figure legend appears to be incomplete, such as "degree, eigenvector, expected influence 1-step, and expected influence 2-step.”

- We thank the reviewer for the attention, and we have corrected the figures’ legend.

Reviewer 3

R package psychNetsAttack is not easy (or impossible?) to access and install (as not so much experienced user I cannot clearly distunguish the latter two; I tried to update my R and RStudio and tried again, unsuccessfully again. BTW I was able to install and implement other (several) R packages). My suggestion is to try (the Editorial Staff or Authors asked by Editors) to load and use the dedicated R package because it is crucial for- and for many readers also important benefit from the reviewed paper. If it works on training dataset then all is OK (so I regret I had no chance to install and launch it on my workstation and my datasets). If installation and/or package is not working, that should be amended before paper is published. Or, the phrases suggesting that every researcher can use the package and repeat reasoning on his/her datasets, should be corrected.

- We agree with the reviewer that there were some difficulties in the install process of the package. This was due to a few packages from which psychNetsAttack package depends on were removed from CRAN. We have already fixed this and installation should work without any problem. If any kind of problem should occur in the installation process, we ask the reviewer to report an issue on GitHub so we can solve it. In line with this, we have also updated the readme file on GitHub) in order to give other researchers the opportunity to use the package (https://github.com/danielcastro86/psychNetsAttack).

The so called 'slim finger' versus 'fat finger' problem is to be discussed in context of influencing nodes in the psychopathological networks in contrast to more technical problems (e.g. net of PCs) or biology (switching off the genes). It is a bit touched with the phrase (322-323): "Do psychotherapeutic strategies and psychopharmacological treatments have the specificity needed to act on a specific central symptom at each time?”

- We would like to thank the reviewer for this highly relevant comment. We have now expanded on this problem in the discussion.

p. 14, line 346 – 353

Although targeting psychological symptoms and behaviors is inherently distinct from targeting genes or computer networks, recent studies [65–70] have started to reveal contrasting effects not only among different treatment modalities [67–70], but also throughout the course of treatment [65,66]. These early results suggest that achieving a remarkable level of precision in psychological treatments may be attainable. This in addition with the selection of treatment targets through network centrality measures has the potential to position psychology on the forefront of precision medicine [71–73] with more effective, precise, and personalized treatment strategies.

Freshly invented centrality measures like hybrid centrality may be mentioned. Same is true for bridge centrality (bridges are discussed).

- We added to our discussion the importance of exploring different centrality measures as, for example, the hybrid centrality.

p. 15, line 364 – 367

Furthermore, there are several unexplored centrality measures that can be explored in the context of psychopathological networks, such as the recently developed hybrid centrality measure [76,77]. Hybrid centrality measures group together several rankings of other centrality measures [77] and can potentially yield valuable insights for the refinement of treatment target selection.

---

## [Decision Letter · Decision Letter 1]

28 Dec 2023

Centrality Measures in Psychological Networks: A Simulation Study on Identifying Effective Treatment Targets

PONE-D-23-24252R1

Dear Dr. Daniel Castro,

We’re pleased to inform you that your manuscript has been judged scientifically suitable for publication and will be formally accepted for publication once it meets all outstanding technical requirements.

Kind regards,

Mu-Hong Chen, M.D., Ph.D.

Academic Editor

PLOS ONE

Additional Editor Comments (optional):

Reviewers' comments:

Reviewer's Responses to Questions

**Comments to the Author**

1. If the authors have adequately addressed your comments raised in a previous round of review and you feel that this manuscript is now acceptable for publication, you may indicate that here to bypass the “Comments to the Author” section, enter your conflict of interest statement in the “Confidential to Editor” section, and submit your "Accept" recommendation.

Reviewer #2: All comments have been addressed

Reviewer #3: All comments have been addressed

2. Is the manuscript technically sound, and do the data support the conclusions?

Reviewer #2: Yes

Reviewer #3: Yes

3. Has the statistical analysis been performed appropriately and rigorously? 

Reviewer #2: Yes

Reviewer #3: Yes

4. Have the authors made all data underlying the findings in their manuscript fully available?

Reviewer #2: Yes

Reviewer #3: Yes

5. Is the manuscript presented in an intelligible fashion and written in standard English?

Reviewer #2: Yes

Reviewer #3: Yes

6. Review Comments to the Author

Reviewer #2: (No Response)

Reviewer #3: All changes needed were made. Thank you for suggestions regarding GitHub.

Congratulations. And all the best for the future.

7. PLOS authors have the option to publish the peer review history of their article (what does this mean?). If published, this will include your full peer review and any attached files.

Reviewer #2: **Yes: **Chih-Wei Hsu

Reviewer #3: No

---

## [Editor Report · Acceptance letter]

3 Jan 2024

PONE-D-23-24252R1 

PLOS ONE

Dear Dr. Castro, 

I'm pleased to inform you that your manuscript has been deemed suitable for publication in PLOS ONE. Congratulations! Your manuscript is now being handed over to our production team.

Kind regards, 

on behalf of

Dr. Mu-Hong Chen 

Academic Editor

PLOS ONE